# Prediction and Measurement of Hovering Flapping Frequency Under Simulated Low-Air-Density and Low-Gravity Conditions

**DOI:** 10.3390/biomimetics10020083

**Published:** 2025-01-29

**Authors:** Hyeonjun Lim, Giheon Ha, Hoon Cheol Park

**Affiliations:** Department of Smart Vehicle Engineering, Future Drone Center, Konkuk University, Seoul 05029, Republic of Korea; dinorca@konkuk.ac.kr (H.L.); hgh6302@gmail.com (G.H.)

**Keywords:** flapping-wing micro-air vehicle, hovering flight, low air density, low gravity, lift coefficient

## Abstract

The ability to predict lift is crucial for enabling flapping flights on planets with varying air densities and gravities. After determining the lift required for a flapping flight on Earth, it can be predicted under different conditions using a scaling equation as a function of air density and gravity, assuming the cycle-average lift coefficient remains constant. However, in flapping wings, passive deformation due to aerodynamic and inertial forces may alter the flapping-wing kinematics, complicating predictions. In this study, we investigated changes in the lift coefficient of flapping wings under various air density and gravity conditions simulated using a low-pressure chamber and tilting stand, respectively. The current study found that the cycle-averaged lift coefficients remained nearly constant, varying by less than 7% across the air density and gravity conditions. The difference between the measured and predicted hovering frequencies increased under a lower air density due to the higher vibration-induced friction. The power consumption analysis demonstrated higher energy demands in thinner atmospheres and predicted a required power of 5.14 W for a hovering flight on Mars, which is a 66% increase compared to that on Earth. Future experiments will test Martian air density and gravity conditions to enable flapping flights on Mars.

## 1. Introduction

Since the first powered flight in 1903 [1], aircraft have advanced significantly. Today, humans explore flights not only on Earth but also on other celestial bodies, including Mars and Titan, which feature distinct air densities and gravitational conditions. For example, Mars has a gravity of approximately 38% of Earth’s, and its air density is only 1.7% of the air density of Earth [2]. Considering gravity and air density conditions, the required speed for flight on Mars is approximately five times greater than that on Earth. Thus, Ingenuity’s rotors rotate at speeds between 2400 and 2700 RPM to achieve flight on Mars [3,4]. On the other hand, the gravity of Titan is approximately 14% of the gravity of Earth [5], and its air density is approximately 45–60% greater than that of Earth [5,6,7,8], providing a more favorable environment for flight. Consequently, the Dragonfly for flight on the Titan [9] was designed to weigh 400–450 kg, compared to Ingenuity’s 1.8 kg [10,11], due to the higher air density and lower gravity conditions on the Titan [12]. Both the Ingenuity and Dragonfly are equipped with rotary wings. Rotary-wing aircraft are capable of vertical and hovering flights and are widely applicable in civil and military fields. Due to their high technological maturity, rotary wings have been selected to design drones for space exploration. In comparison, flapping-wing technology remains less mature despite recent progress [13,14,15,16,17,18,19,20,21]. Therefore, flapping wings have not yet been selected for space drone design.

Flapping wings offer several advantages over rotary wings. Flapping-wing aircraft, unlike rotary-wing aircraft [22], do not require anti-torque mechanisms produced by a rotor system because the torques generated by a pair of wings are canceled out, and theoretically, no yaw moment is developed. Moreover, due to the symmetric inflow distributions in the left and right wings, no rolling moment is produced, even during forward flight. Thus, flapping wings are theoretically more stable during forward flight. Additionally, the flapping wings create passive deformation to produce the required lift during the upstroke and downstroke. To achieve this, flapping wings are constructed with a stiff leading edge and a thin, flexible plastic film reinforced by artificial vein structures. Therefore, wing fabrication is relatively inexpensive and economical. Compared with flapping wings, the rotor of a rotary-wing aircraft is difficult and expensive to manufacture due to its required shapes for aerodynamics and materials for fabrication. Thus, flapping wings are an alternative drone design option for space [23,24]. A tailless insect-like flapping-wing aircraft is particularly suited for future space exploration because it is hover-capable and compact in shape without control surfaces at the tail, which is beneficial for packing.

Tailless flapping-wing aircraft have been studied in many laboratories and are comprehensively reviewed in [13,14]. This study focuses specifically on those that have successfully demonstrated sustained, controlled flight with onboard controls and power sources. The Nano Hummingbirds demonstrated the first successful controlled flight, which mimicked the flapping wing motions of hummingbirds and flew for 4 min. Saturn, a nano hummingbird without a fairing structure, demonstrated an endurance of approximately 11 min [25]. Delfly Nimble, developed by the Delft University of Technology, is a robot that can perform agile maneuvers that mimic the bank turn of a fruit fly. It has an endurance of approximately 5 min and a forward flight speed of 7 m/s [16]. NUS-robotic birds also do not have control surfaces on their tails but can perform quick turns flying for approximately 3.5 min [26]. The Purdue Hummingbird is another tailless flying robot based on a resonant flapping mechanism powered by an onboard battery [27], whose wings are independently actuated, similar to the Delfly Nimble. We also developed a two-winged tailless flapping-wing aircraft and robot [28,29,30,31]. KUBeetle achieved its first controlled flight [31], KUBeetle-S recorded a flight time of approximately 9 min [30], and KUBeetle-RP performed a controlled flight for approximately 7 min [32]. Recently, we developed collapsible wings that mimic the collision energy absorption of real beetle hindwings. When KUBeetle is equipped with collapsible wings, it can maintain stable flight even after the wing tip collides with an obstacle [28]. Thus, collapsible wings may enhance the likelihood of completing the missions assigned to flapping-wing aircraft. Furthermore, similar to beetles, another version of KUBeetle demonstrated passive wing folding and unfolding and performed controlled flight with passively foldable wings [29]. If the wings of a flapping-wing vehicle can be folded, it can be packed in a smaller volume and easier to transport from one place to another, for example, from Earth to Mars. Therefore, the technology for flapping wings is becoming more mature.

However, research on flapping wings under low air densities is still in its early stages. The takeoff of a 12.95-g flapper under 33% air density at sea level on Earth was first reported in [33], where resonant-type two-winged flappers with no control were used for the demonstration. Changes in flapping characteristics under low-air-density conditions have been studied with a dynamic model and experimentally in [34], where we found that for the same input voltage, the flapping frequency can be increased by more than 10% when the air density decreases to 10% of the air density at sea level because of lower drag. However, power consumption tends to increase to overcome the larger inertial torques. An increase in the inertial power of flapping wings under Martian conditions was also reported in [24,34]. Based on the assumption that wing kinematics are not altered even for flapping flight under Martian air density conditions, which require approximately five times higher flapping frequency, we also found that the stability characteristics of hovering KUBeetles on Mars are similar to those on Earth. In the calculation of the required flapping frequency for flight on Mars, we assumed that the lift coefficient did not change, even for faster flapping wing motions. However, this assumption has not been experimentally proven.

This study investigates the lift-generation characteristics of a flapping-wing model under conditions of low air density and gravity. The flapping mechanism based on the rack-pinion, which was adopted in the KUBeetle-RP [32], was used to create a flapping flight model without a control system or battery to meet the weight requirement for takeoff under a low air density of up to 25% of the air density at sea level and low gravity conditions. Low air density was simulated in the same low-pressure chamber used in [34], and the low-gravity condition was represented by appropriately tilting the body axis of the flapping-wing model instead of using the weight compensation technique [35]. Hovering flapping frequencies of the flapping-wing model were measured under simulated low-air-density and low-gravity conditions, and the measured hovering flapping frequencies were compared with the predicted frequencies using a scaling equation. The scaling equation was derived from a simple lift equation based on the assumption that the lift coefficient does not change under low-air and low-gravity conditions. By comparing the measured and predicted frequencies, we investigated the changes in the cycle-average lift coefficient of the flapping-wing model under low-air-density and low-gravity conditions.

Section 2 describes the experimental simulation of low-air-density and low-gravity conditions. Details of how the flapping-wing model was constructed and how the experiments were performed are also included. Section 3 summarizes and compares the required lift and measured hovering flapping frequency for each set of air density and gravity conditions, the measured lift at each measured flapping frequency, and the computed lift coefficients. Thus, we attempted to prove the effectiveness of the proposed experimental setup and the assumption of a constant lift coefficient for flight under a low air density, as discussed in Section 4. Finally, Section 5 concludes the paper and presents future research plans.

## 2. Materials and Methods

### 2.1. Calculation of the Predicted Frequency and Lift Coefficient

This section presents a scaling equation to predict the flapping frequency required for a flapping-wing model to hover under varying air density and gravity conditions. Because the wings need to flap faster to generate lift under low air density, the maximum achievable flapping frequency determines how low the air density is for the flapping-wing aircraft to fly. Therefore, it is essential to predict and verify flapping frequencies. To derive the scaling equation, lift (L) in Equation (1) [36] as follows:
(1)$$L=0.5\times \rho \times C_{L}\times V^{2}\times S,$$
where ρ is the air density, $C_{L}$ is the lift coefficient, V is the reference speed, and S is the wing area. The scaling equation for predicting the flapping frequency is derived as follows: first, Equation (1) can be rewritten as Equation (2) by expressing the reference speed (V) in terms of the flapping frequency f, the flapping amplitude ψ, and the one-wing length b.
(2)$$L=0.5\times \rho \times C_{L}\times {\left(\frac{\pi b\psi }{180}\right)}^{2}\times {\;f\;}^{2}\times S$$
Here, the $C_{L}$ is the cycle-average lift coefficient. If the cycle-average lift coefficient remains constant across air density and gravitational acceleration, the flapping frequency required for hovering is inversely proportional to the square root of the air density and proportional to the square root of the gravitational acceleration. The ratio between the required lift on Earth and another planet is expressed as follows:
(3)$$\frac{L_{A}}{L_{E}}=\frac{g_{A}}{g_{E}}=\frac{\rho_{A}}{\rho_{E}}{\left(\frac{V_{A}}{V_{E}}\right)}^{2}=\frac{\rho_{A}}{\rho_{E}}{\left(\frac{f_{A}}{f_{E}}\right)}^{2},$$
where the subscript *E* and *A* stand for Earth and another planet, respectively, and *g* means the gravity acceleration. Therefore, the scaling equation that predicts the flapping frequency required for hovering flight under specific air density and gravity conditions can be expressed by Equation (4)
(4)$$f_{A}=\sqrt{\left(\rho_{E}\times g_{A})/(\rho_{A}\times g_{E}\right)}\;\times \;f_{E}$$

From Equation (4), it can be confirmed that a flapping-wing aircraft requires a faster flapping frequency as the air density decreases and can achieve flight with a slower flapping frequency as gravity decreases. After measuring the flapping frequency under a given set of air densities and gravity conditions, the measured flapping frequency was compared with the value predicted using Equation (4). If the two frequencies are close, the cycle-average lift coefficient is considered constant.

### 2.2. Experiment

#### 2.2.1. The Flapping-Wing System

The flapping mechanism (Figure 1A) in the flapping-wing model used in this experiment is structurally identical to the rack-pinion mechanism described in [34]. The flapping-wing model (Figure 1B) was powered by a single direct current (DC) motor (CL0720-14; Chaoil, Guangdong, China) coupled with a set of reduction gears with a gear ratio of 30.5:1.

The wings used in this study had the same structure and construction as those used in [29]. The area of one wing is 25.5 $cm^{2}$, the aspect ratio is 3.2, and the length of one wing is 9 cm, respectively. Three 0.25 mm thick carbon rods served as veins and stiffened the wing membrane. The membrane is made of 10 μm thick Mylar film. To prevent excessive bending of the leading edge at high flapping frequencies, a thicker leading edge (1.5 mm) was used than in previous studies [28,30]. The total mass of the flapping-wing model, including the upper and lower guides (Figure 1B), was 11.70 g.

#### 2.2.2. Low-Air-Density Condition

A chamber with adjustable pressure was used to simulate a low-density atmospheric environment [37]. The chamber ($0.73\;m^{3}$) was the same as that used in [34]. The chamber was made of 15 mm thick polycarbonate (PC) and aluminum profiles ($40\;mm^{2}$) were installed inside to protect the PC walls. In this study, a flapping-wing flight experiment was conducted under the conditions of 25%, 50%, and 100% air density at sea level (1.225 kg/m^3^) [38]. The potential error in the accuracy of air density can be ±1.5% because it is the accuracy of the pressure gauge (WS-110-ϕ60, WooShin, Bucheon, Republic of Korea).

#### 2.2.3. Low-Gravity Condition

The low-gravity environment was simulated using a stand that can tilt the installed flapping-wing model at a specific angle θ with respect to the horizontal line, as shown in Figure 2. When the flapping-wing model is tilted using the stand, the gravitational acceleration opposite to the flight direction is reduced to gsin θ. The tilt angles (θ) used in this study are 90°, 45°, 30°, and 22°. Tilting angles of 90° and 22° were used to simulate the gravitational acceleration on Earth (1 g) and Mars (0.38 g), respectively. The intermediate angles of 45° and 30° were chosen to simulate further different gravitational accelerations of 0.71 g and 0.50 g, respectively.

The stand used in this study consisted of two main parts, as shown in Figure 2 and Figure 3: a rotating frame, where the flapping-wing model was mounted, and fixed frames that supported the rotating frame. The stand was constructed using lightweight and durable materials, including aluminum profiles, carbon plates, transparent acrylic plates, and 3D-printed parts. A servomotor (D955TW, HITEC, Seoul, Republic of Korea) was used to tilt the rotating frame, and it was controlled using an Arduino UNO (Arduino, Scarmagno, Italy). A Bluetooth module (HC-06, 2.4 GHz ISM band frequency, Guangzhou HC Information Technology Co., Ltd, Guangzhou, China) was used to control the tilting angle from outside the chamber remotely. The stand was positioned inside the low-air-density chamber. The flapping-wing model was installed at a height sufficient to mitigate ground effects (>three times the length of one wing) [39]. The distance between the wing tip and the wall of the chamber was sufficiently large to ignore the wall effect (>2.4 times the one wing length). We measured the flapping frequency of the flapping-wing model when it took off and stayed in one position, that is, hovering.

The vertical movement of the flapping-wing model must be smooth to accurately simulate vertical takeoff under a given gravity condition or tilting angle. Therefore, two sets of bearings (upper: 22ϕ, 7 mm, JIS class 0, MISUMI, Tokyo, Japan, lower: 17ϕ, 7 mm, JESA, Villars-sur-Glâne, Switzerland) were installed to hold the two guides, which were attached at the top and bottom of the flapping-wing model, as shown in Figure 3. With this design, the flapping-wing model could take off smoothly in the flight direction.

It is essential to minimize the blocking of airflow by rotating frames because the flapping-wing model takes off and hovers between the upper and lower frames. Therefore, the upper and lower frames were made in the form of a grid structure consisting of two thin carbon plates, as shown in Figure 4, to prevent the blocking of the airflow from the top of the flapping-wing model to the bottom.

#### 2.2.4. Verification of the Stand

To verify that the stand could simulate low-gravity conditions, we measured the lifts produced at the measured hovering frequencies. The hovering flapping frequencies were measured when the rotating frame was tilted from 90° (Earth gravity condition) to 45°, 30°, and 22° (Martian gravity condition) at 100% air density. Then, the lift of the flapping-wing model was measured at each measured hovering flapping frequency by using a load cell (Nano 17, ATI Industrial Automation, Apex, NC, USA, force resolution ≈ 0.32 gf). The results, shown in Table 1, indicate no significant difference between the predicted lift by mgsin θ and measured lifts for the four tilting or gravity conditions. These findings confirm that the stand could effectively simulate various gravity conditions. Therefore, the blocking of the airflow by the upper and lower rotating frames was negligible. In addition, the results imply that the friction caused by vibration and gravity is small enough to ignore when the flapping frequency is not very high. In Table 1, “Tilting test (HSC)” means the flapping frequency measured by using a high-speed camera (FASTCAM Ultima APX, Photron, Tokyo, Japan), and “Load cell (Signal)” stands for the frequency processed from the force signal sensed by the load cell. The measurement data using the load cell in Table 1 were acquired by taking an average of three measurements.

#### 2.2.5. Experiment Setup

In this study, the hovering flapping frequency of a flapping-wing model was measured under three air densities and four gravitational conditions. The air densities were set to 100% (sea level on Earth), 50%, and 25%. Gravitational conditions were simulated using tilt angles of 90° (gravity of Earth, 1 g), 45° (0.71 g), 30° (0.5 g), and 22° (0.38 g, simulating Martian gravity) as previously mentioned. As the flapping-wing model could not take off under air density conditions lower than 25%, we did not experiment with air densities lower than 25%. Figure 5 shows the experimental setup. The chamber pressure was reduced to simulate a low-air-density condition, and the rotating frame was tilted at an angle to reproduce the gravity condition. A power supply (E36103A, Keysight, Santa Rosa, CA, USA) was used to apply voltage to initiate the takeoff of the flapping-wing model, and the voltage was gradually reduced until the flapping-wing model reached a state of hovering after takeoff. The flapping frequency at this point is recorded and regarded as the hovering flapping frequency.

The power consumption of the flapping-wing model was measured at each hovering flapping frequency for a set of low-air-density and low-gravity conditions. The power-measurement setup involved resistors, a power supply, and an oscilloscope (TDS 2024; Tektronix, Beaverton, OR, USA), as shown in Figure 6. The oscilloscope measured the output voltage of the power supply (V_1_), and the input voltage applied to the flapping-wing model (V_2_). Since the currents (I) applied to the resistor and flapping-wing model are the same [40], the current can be calculated as I = (V_1_ − V_2_)/R_1_. Due to limitations in measuring the voltage applied to the motor (V_3_) inside the chamber, the resistance of the wire (R_2_) in the chamber was measured. Then, the voltage across the motor was calculated with V_3_ = V_2_ − IR_2_. Subsequently, the power consumption of the flapping-wing model can be calculated by multiplying the voltage V_3_ by the current I.

## 3. Results

### 3.1. Hovering Test

Figure 7 and Figure 8 show that the hovering flight of the flapping-wing model was achieved under all gravity conditions when the air density was 100% and 50%, respectively. However, at 25% air density (Figure 9), the flapping-wing model could not take off under the gravity of Earth (90°) due to insufficient lift generation in a thinner atmosphere, as shown in Figure 9A. The initial level in each figure indicates the position at which the flapping-wing model starts to take off, and the hovering level indicates the hovering state of the flapping-wing model. Appendix A demonstrating the hovering flight under various conditions are provided as Appendix A.

### 3.2. Measured Flapping Frequency

The measured hovering flapping frequency, predicted frequency, and cycle-average lift coefficient calculated based on the measured flapping frequency under different air densities and gravity conditions are summarized in Table 2, Table 3 and Table 4. In the first column of each table, the tilt angle simulates varying gravitational accelerations, as previously described. The lift in the second column lists the required lift for each gravity condition, which is computed with mgsin θ.

The predicted frequencies in Table 2, Table 3 and Table 4 were computed with scaling Equation (3), based on the measured hovering flapping frequency for 100% air density and 1 g (90°) conditions as the reference frequency. The cycle-average lift coefficient was calculated using Equation (2). The percentages in parentheses indicate the difference between the reference cycle-average lift coefficient for the 100% air density and 1 g (90°) condition and that for the other conditions.

Table 2, Table 3 and Table 4 demonstrate that the hovering flapping frequency increases as air density decreases and it decreases for a smaller tilting angle, which stands for lower gravity in the flight direction. Thus, the results match these trends. At 100% air density, as shown in Table 2, the measured hovering flight frequency closely matched the predicted values, with a difference of less than 1% across all gravity conditions. The cycle-average lift coefficient remained consistent at approximately 1.02 under all gravity conditions. According to Table 3, the measured frequencies at 50% air density were, on average, 2.2% lower than the predicted frequencies. The cycle-average lift coefficient is approximately 1.03, which is an increase of approximately 1.3% compared with that for 100% air density. The results at 25% air density in Table 4 indicate the highest measured flapping frequency for hovering flights. However, due to the significantly reduced air density, the generated lift was insufficient for takeoff at a tilting angle of 90° (1 g), even when we applied voltages higher than the nominal voltage to the motor, as observed in Figure 9A. The average difference between the measured and predicted frequencies was approximately 5.0%, which was the highest among the three air density conditions. The cycle-average lift coefficient remained consistent across all tilting angles and gravity conditions at approximately 1.08, which is an increase of approximately 6.2% compared with 100% air density. At higher flapping frequencies under lower air densities (Table 3 and Table 4), more vibration of the flapping-wing model may cause high friction between the rods and bearings, which results in an increase in the difference between measured and predicted hovering frequencies. In the current case, the friction force supports the flapping-wing model such that it can hover at a lower flapping frequency, which results in a slightly higher cycle-average lift coefficient.

### 3.3. Power Consumption

As listed in Table 5, the power consumption was measured for 11 cases, excluding the tilting angle of 90° (1 g) and 25% air density, where the flapping-wing model could not take off. For 100% and 50% air densities, when the tilting angle of the flapping-wing model was changed from 90° (1 g) to 45° (0.71 g) and from 45° (0.71 g) to 30° (0.5 g), respectively, the power consumption consistently decreased by approximately 40%. When the tilt changed from 30° (0.5 g) to 22° (0.38 g, simulating Martian gravity), the reduction rate was 30%. In the case of 25% air density, when the tilting angle changed from 45° to 30° and from 30° to 22°, the power consumption decreased by 54% and 38%, respectively. This reduction was primarily attributed to the lower flapping frequency required under low-gravity conditions.

Figure 10A shows that the power consumption increases as the air density decreases to generate the same amount of lift because a higher flapping frequency is required to compensate for the lower air density. Consequently, it can be inferred that flapping-wing aircraft will continue to demand greater power for flight in low-density atmospheres. However, as illustrated in Figure 10B, less power is required to generate the same flapping frequency as the air density decreases. This suggests that for a lower air density, the air resistance diminishes so that a higher flapping frequency can be achieved by consuming relatively lower power. Figure 10B further demonstrates that by halving the air density, the power required to generate a similar flapping frequency did not decrease by half. This implies that the aerodynamic force contributes a relatively small portion of the total power consumption, which aligns with the findings of previous studies [34]. Thus, the results suggest that the current flapping-wing model consumes significant power to overcome inertial forces instead of aerodynamic drag under 25% air density. Using these experimental results, the power required for the flapping-wing model to operate under Martian surface conditions was predicted. The prediction methodology and findings are discussed in detail in the Section 4.

## 4. Discussion

Hovering at lower air densities requires a higher flapping frequency, which increases vibrations and, consequently, friction. In the current experiments, higher friction facilitated hovering flight so that the flapping-wing model could take off at a lower frequency. Because of the lower measured frequency, the cycle-average lift coefficient increased slightly, as shown in Table 3 and Table 4. The cycle-average lift coefficients were nearly identical under the same air density conditions across the four gravity conditions (Table 2, Table 3 and Table 4). This indicates that tilting the flapping-wing model is effective for simulating low-gravity conditions. For a more accurate assessment of the cycle-average lift coefficient, we may need to estimate the contribution of friction to the lift coefficient further.

Figure 11A presents a graph interpolated from the 11 measured data points in Table 2, Table 3 and Table 4, illustrating the changes in power consumption under different air and gravity conditions. The graph indicates that, as the air density decreases, lift generation becomes more challenging due to increased power consumption. As gravity decreases, the lift requirement for flight diminishes, leading to lower power consumption.

While the flapping frequency can be reliably predicted using the scaling equation detailed in Section 2.1, predicting the power consumption is more complex. This complexity arises because the power consumption to overcome the inertial force increases [34], although the power consumption due to aerodynamic forces decreases for a lower air density.

Figure 11B presents measured power data under various gravity conditions, fitted with a logarithmic function using the least-squares method for each air density. The logarithmic function was selected because it appropriately models theoretical behavior in which the power consumption becomes infinite as the air density approaches zero, reflecting an impossible flight without air. These fitting functions enable the prediction of power consumption for arbitrary combinations of air density and gravity conditions. Figure 11B also indicates that if the flight characteristics of the flapping-wing model remain consistent in the Martian environment, the power required for its hovering flight can be estimated by the fitting function (power = −1.059 ln (air density ratio) + 0.83). When the Martian air density ratio of 0.017 is substituted into the fitting function, the predicted power is approximately 5.14 W. The flapping frequency necessary is predicted as 68.7 Hz for flight on Mars, according to the scaling Equation (3). Thus, approximately 66% more power is required to create a flapping frequency that is approximately five times higher on Mars than on Earth. If we can add more measured power consumptions for lower air densities of 5% and 10% air density in the fitting curve, the power consumption for flapping flight on Mars can be more accurately predicted.

## 5. Conclusions and Future Work

The proposed tilting-angle-adjustable stand effectively simulated a low-gravity environment, while the low-air-density chamber correctly reproduced low-density conditions. Under simulated low-density and low-gravity conditions, the 11.7 g flapping-wing model successfully achieved hovering flight, with its hovering flapping frequency and power consumption measured.

The analysis revealed consistent differences between the measured and predicted frequencies for each air density. These differences increased as the air density decreased. This discrepancy arose because of the increase in friction due to the higher vibration caused by a higher flapping frequency under lower air density conditions. In this study, higher friction supported hovering flight, allowing the flapping-wing model to hover at a lower frequency than predicted.

Similarly, the cycle-average lift coefficient remained consistent across the three air densities for all tilt angles but increased by approximately 6% as air density decreased to 25% due to the lower measured hovering frequency compared to predictions. Thus, the cycle-average lift coefficient appears unaffected mainly, even at 25% air density.

Decreasing air density requires the flapping-wing model to flap faster to achieve flight, resulting in higher power consumption. However, the reduced drag under low-air-density conditions allows faster wing flapping at the same power level. From the power consumption data collected under various air density and gravity conditions, it is estimated that the power consumption required for a hovering flight on Mars is approximately 5.14 W, which is 66% higher than the required for a hovering flight on Earth if the flight characteristics of the current flapping-wing model remain consistent under Martian conditions.

To more accurately measure the hovering flapping frequency and calculate the cycle-average lift coefficient, we will suggest a way to reduce the effect of friction at higher flapping frequencies under low-air-density and low-gravity conditions in the future. We also plan to demonstrate the hovering flapping flight of a flapping-wing model at lower air density conditions, which is closer to the Martian air density, and measure the corresponding power consumption for a more accurate prediction of the power requirement for flapping flight on Mars.

## Figures and Tables

**Figure 1 biomimetics-10-00083-f001:**
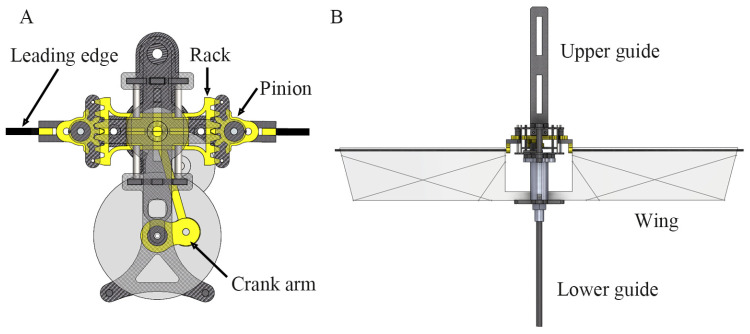
(**A**) Flapping mechanism based on the rack-pinion system, (**B**) flapping-wing model with guides.

**Figure 2 biomimetics-10-00083-f002:**
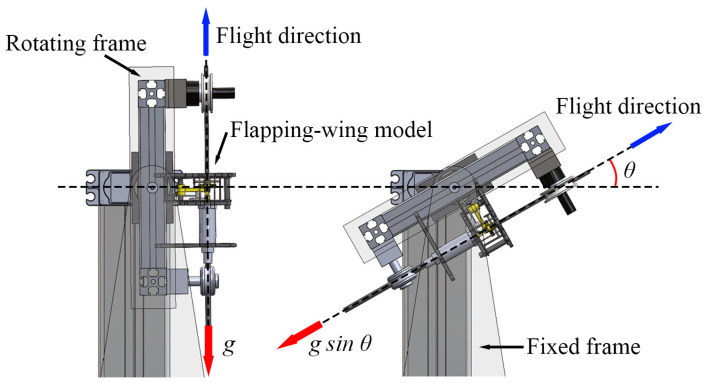
Appearance of the stand and the flapping-wing model mounted on it tilting.

**Figure 3 biomimetics-10-00083-f003:**
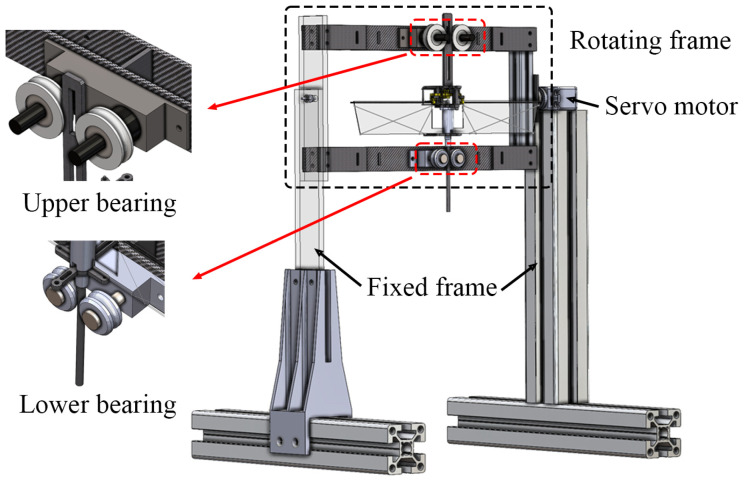
The stand configuration and the flapping-wing model mounted on the stand.

**Figure 4 biomimetics-10-00083-f004:**
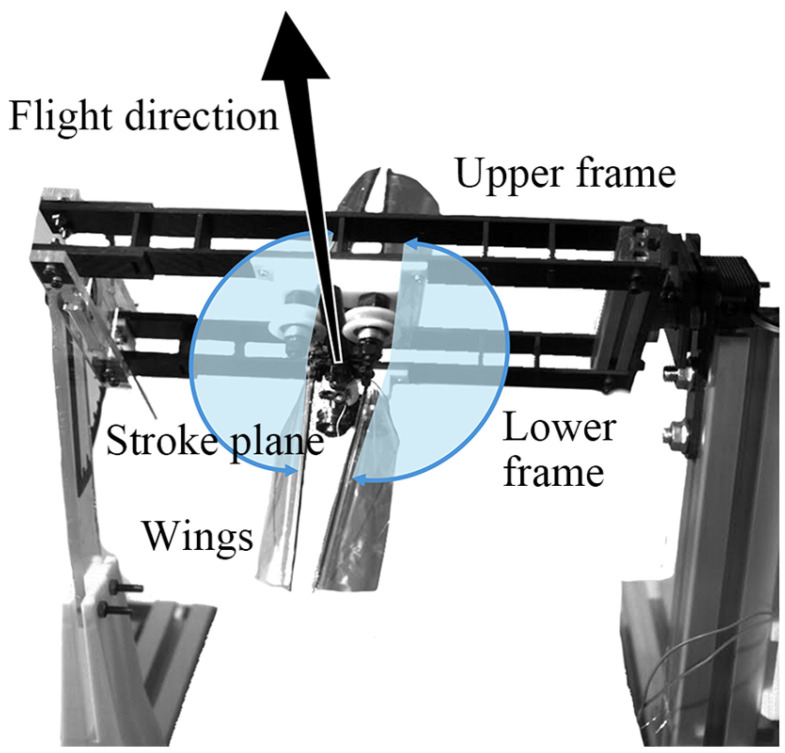
Actual view of the flapping-wing model installed to the rotating frame.

**Figure 5 biomimetics-10-00083-f005:**
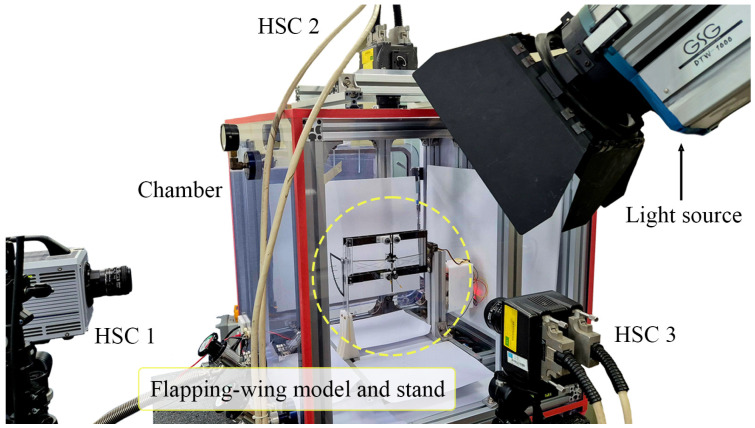
Experimental setup for measuring flapping frequency and amplitude of flapping-wing model: the stand with the flapping-wing model was installed inside the low-air-density chamber, and three high-speed cameras were used for filming.

**Figure 6 biomimetics-10-00083-f006:**
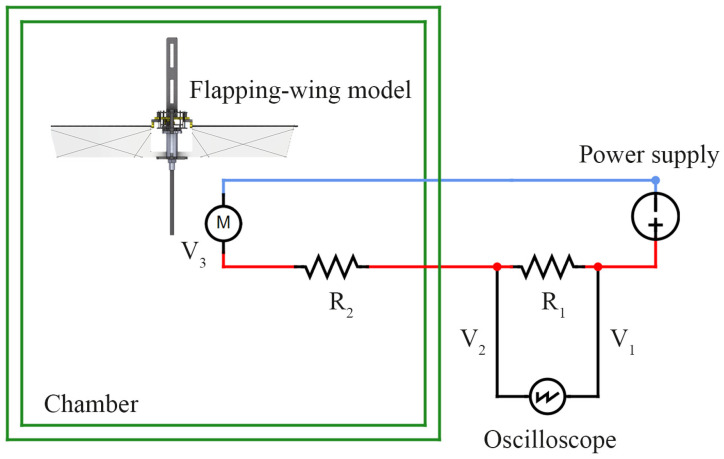
The circuit diagram for measuring the power consumption of the flapping-wing model.

**Figure 7 biomimetics-10-00083-f007:**
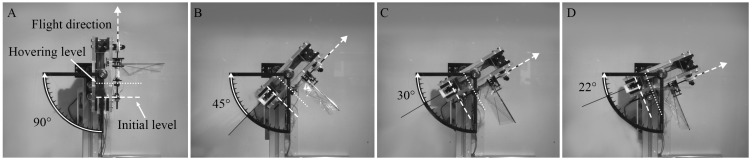
Hovering of the flapping-wing model at 100% air density under the four different gravity conditions: (**A**) gravity acceleration of Earth (g), (**B**) 0.71 g, (**C**) 0.5 g, (**D**) Martian condition (0.38 g).

**Figure 8 biomimetics-10-00083-f008:**
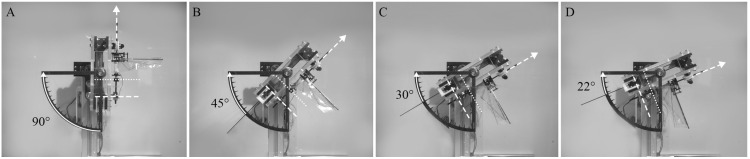
Hovering of the flapping-wing model at 50% air density under the four different gravity conditions: (**A**) gravity acceleration of Earth (g), (**B**) 0.71 g, (**C**) 0.5 g, (**D**) Martian condition (0.38 g).

**Figure 9 biomimetics-10-00083-f009:**
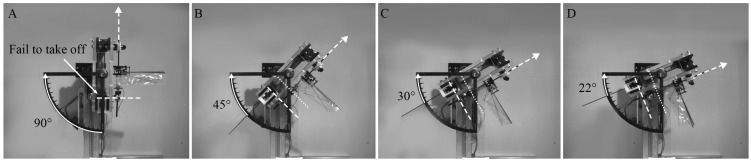
Hovering of the flapping-wing model at 25% air density under the four different gravity conditions: (**A**) gravity acceleration of Earth (g), (**B**) 0.71 g, (**C**) 0.5 g, (**D**) Martian condition (0.38 g).

**Figure 10 biomimetics-10-00083-f010:**
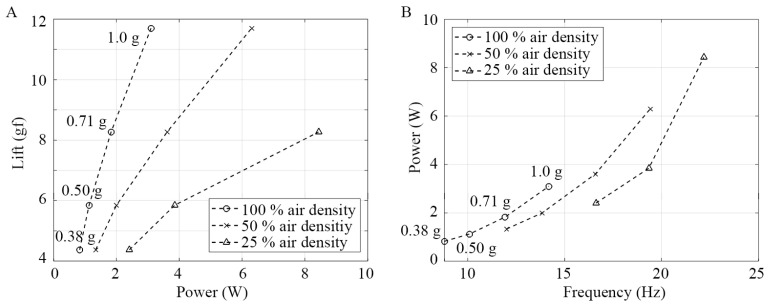
(**A**) Measured lift versus power consumption, (**B**) measured flapping frequency versus power consumption.

**Figure 11 biomimetics-10-00083-f011:**
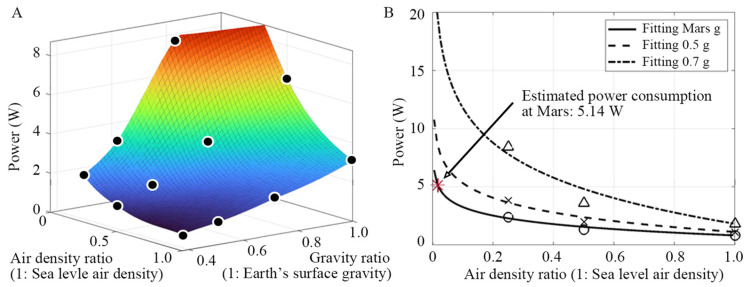
(**A**) 3D graph of power consumption measured during hovering flight under various air densities and gravity conditions. (**B**) Results of fitting the measured power consumption according to air density by gravity condition using the log function. The circle, cross, and triangle symbols represent measured power data points under different gravity conditions.

**Table 1 biomimetics-10-00083-t001:** A comparison of the lift was calculated according to the inclination of the flapping-wing model, and the lift was measured by the load cell.

Lift (gf)		Frequency (Hz)	
Prediction (=mgsin θ)	Measurement (Load cell)	Tilting test (HSC)	Load cell (Signal)
11.70 (θ=90°)	11.97 (+2.35%) (SD = 0.0449)	14.21	14.16 (−0.35%) (SD = 0.0264)
8.27 (θ=45°)	8.49 (+2.69%) (SD = 0.0410)	11.95	11.98 (+0.25%) (SD = 0.0273)
5.85 (θ=30°)	5.98 (+2.26%) (SD = 0.0595)	10.07	10.08 (+0.10%) (SD = 0.0105)
4.38 (θ=22°)	4.49 (+2.61%) (SD = 0.0282)	8.77	8.72 (+0.57%) (SD = 0.0182)

**Table 2 biomimetics-10-00083-t002:** Experiment results of tilting test at 100% air density.

Tilting Angle	Lift (gf) (mgsin θ)	Measured Freq. (Hz)	Predicted Freq. (Hz)	Difference (%)	Cycle-Average Lift Coefficient
90° (1.00 g)	11.70	14.21	14.21 (Ref.)	N/A	1.022 (Ref.)
45° (0.71 g)	8.27	11.95	11.95	+0.02	1.021 (+0.10)
30° (0.50 g)	5.85	10.07	10.05	+0.25	1.027 (+0.49)
22° (0.38 g)	4.38	8.77	8.69	+0.90	1.025 (+0.29)

**Table 3 biomimetics-10-00083-t003:** Experiment results of tilting test at 50% air density.

Tilting Angle	Lift (gf) (mgsin θ)	Measured Freq. (Hz)	Predicted Freq. (Hz)	Difference (%)	Cycle-Average Lift Coefficient
90° (1.00 g)	11.70	19.57	20.09	−2.59	1.034 (+1.17)
45° (0.71 g)	8.27	16.52	16.89	−2.15	1.034 (+1.17)
30° (0.50 g)	5.85	13.87	14.21	−2.36	1.039 (+1.66)
22° (0.38 g)	4.38	12.09	12.30	−1.65	1.034 (+1.17)

**Table 4 biomimetics-10-00083-t004:** Experiment results of tilting test at 25% air density.

Tilting Angle	Lift (gf) (mgsin θ)	Measured Freq. (Hz)	Predicted Freq. (Hz)	Difference (%)	Cycle-Average Lift Coefficient
90° (1.00 g)	11.70	N/A	28.41	N/A	N/A
45° (0.71 g)	8.27	22.37	23.89	−6.36	1.084 (+6.07)
30° (0.50 g)	5.85	19.23	20.09	−4.27	1.081 (+5.77)
22° (0.38 g)	4.38	16.56	17.39	−4.51	1.091 (+6.75)

Note. The percentages in parentheses represent the relative differences (%) between the quantities for reference (Ref. in Table 2) and other conditions.

**Table 5 biomimetics-10-00083-t005:** Power consumption occurs when hovering across the air density and gravity conditions.

Power (W)	Air Density Condition		
Gravity condition	100%	50%	25%
90° (1.00 g)	3.10	6.30	N/A
45° (0.71 g)	1.83 (−41.0%)	3.61 (−42.7%)	8.43
30° (0.50 g)	1.13 (−38.4%)	2.00 (−44.5%)	3.85 (−54.3%)
22° (0.38 g)	0.83 (−26.8%)	1.34 (−33.1%)	2.41 (−37.4%)

## Data Availability

All data measured or analyzed during this study were included in this published article.

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
