# Peer review of "Prediction and Measurement of Hovering Flapping Frequency Under Simulated Low-Air-Density and Low-Gravity Conditions"

_biomimetics, 2025, doi:10.3390/biomimetics10020083_

Round 1
Reviewer 1 Report
Comments and Suggestions for Authors
The manuscript presents an interesting approach to simulate the hovering flight of a bio-inspired flapping-wing vehicle under Mars-like conditions by modifying the environmental pressure and adjusting the tilt angles of the aircraft to simulate low-density and gravity environments. The experimental results detail the differences produced by these variables and predict the power requirements for achieving hovering flight on Mars. However, I have concerns that the manuscript shares a significant amount of similarity with reference [29]. I would like the authors to provide a detailed explanation of what new findings have been obtained in this study based on the foundation of reference [29].
The following are other concerns that need to be addressed by the authors:
- In Figure 2, does the tilted test rig effectively simulate a low-gravity environment? The pressure exerted by the gravity component on the rod seems to generate additional friction. The authors should clarify if the frictional forces from the tilted setup are fully accounted for in their simulation of low gravity.
- In Figure 3, does the scale of the aircraft relative to the test rig accurately reflect the experimental setup? The large size of the vehicle, combined with the wing flapping, is likely to create airflow disturbances around the rig that could affect the results. Please clarify how such effects were controlled or accounted for in the experiment.
- In line 252, the authors refer to the power calculation in reference [29], yet Figure 6 is presented to demonstrate the power calculation method. This is somewhat confusing. I suggest either providing a detailed description of the power calculation method in the manuscript or removing Figure 6 to avoid redundancy.
- In lines 310-313, the authors mention that friction is primarily caused by higher flapping frequencies. However, it is important to note that additional friction may also result from the gravity component acting on the test setup. The authors should clarify whether gravity-induced friction was taken into account and if it might have impacted the results.
- In line 326, Table 1 appears again. Please correct the numbering of the tables to avoid confusion. Additionally, it is unclear whether the values in Table 1 represent averages across multiple trials or the results from a single experiment. Please clarify this point.
- All the supplementary videos currently provided are based on a purely lateral view, which makes it difficult to discern the differences in the experimental results. I recommend the following improvements:
- Provide additional views to capture different angles of the experiment.
- Use tracer particles to visualize airflow and flight patterns more effectively.
- Synchronize the presentation of experimental data for improved clarity and better demonstration of the effects observed in the trials.
Author Response
Reply to the reviewers' comments
Reviewer #1
We appreciate the Reviewer’s comments. All the comments were very useful to improve the quality of our paper. We have revised the manuscript reflecting most of the comments and hope that this revision fits the publication standards of Biomimetics.
General comments: The manuscript presents an interesting approach to simulate the hovering flight of a bio-inspired flapping-wing vehicle under Mars-like conditions by modifying the environmental pressure and adjusting the tilt angles of the aircraft to simulate low-density and gravity environments. The experimental results detail the differences produced by these variables and predict the power requirements for achieving hovering flight on Mars. However, I have concerns that the manuscript shares a significant amount of similarity with reference [29]. I would like the authors to provide a detailed explanation of what new findings have been obtained in this study based on the foundation of reference [29].
Response: The work in [29] is very different from the current research. As already mentioned in Introduction, in [29], the same voltages were applied to the flapping-wing model for all air density conditions to see how the flapping frequency and motion change in the low air density conditions for the same voltage input. On the other hand, in this study, we applied various voltages to make the flapping-wing model hover under low air densities and gravity conditions. Therefore, while the study in [29] focused on the change in wing flapping characteristics at low air density when the same voltage was applied, in this study, we investigated the change in wing flapping frequency and cycle-average lift coefficient when the flapping-wing model hovers under low air density and gravity conditions.
Comments 1: In Figure 2, does the tilted test rig effectively simulate a low-gravity environment? The pressure exerted by the gravity component on the rod seems to generate additional friction. The authors should clarify if the frictional forces from the tilted setup are fully accounted for in their simulation of low gravity.
Response 1: The effectiveness was verified in 2.2.4, even though we did not describe the effect of friction in that section. The effect of friction is included in Discussion. So, in the revised manuscript, we added the following sentence in 2.2.4.
(See lines 226 to 229 on page 7 in the revised manuscript.)
(Before)
These findings confirm that the stand could effectively simulate various gravity conditions. Additionally, the blocking of the airflow by the upper and lower rotating frames was negligible.
(After)
These findings confirm that the stand could effectively simulate various gravity conditions. Therefore, the blocking of the airflow by the upper and lower rotating frames was negligible. In addition, the results imply that the friction caused by vibration and gravity is small enough to ignore when the flapping frequency is not very high.
Comments 2: In Figure 3, does the scale of the aircraft relative to the test rig accurately reflect the experimental setup? The large size of the vehicle, combined with the wing flapping, is likely to create airflow disturbances around the rig that could affect the results. Please clarify how such effects were controlled or accounted for in the experiment.
Response 2: Figure 3 displays the CAD design of the actual test setup. The effect of air flow disturbance as well as the friction can be negligible, as already claimed in 2.2.4, because each measured lift in the inclined direction is close to corresponding required lift, which is in Table 1.
Comments 3: In line 252, the authors refer to the power calculation in reference [29], yet Figure 6 is presented to demonstrate the power calculation method. This is somewhat confusing. I suggest either providing a detailed description of the power calculation method in the manuscript or removing Figure 6 to avoid redundancy.
Response 3: Reflecting the reviewer’s comment, we included Details on the power measurement method are included in the revised manuscript.
(See lines 257 to 264 on page 8 in revised manuscript)
The power consumption of the flapping-wing model was measured at each hovering flapping frequency for a set of low-air-density and low-gravity conditions. The power measurement setup involved resistors, a power supply, and an oscilloscope (TDS 2024; Tektronix, USA), as shown in Figure 6. The oscilloscope measured the output voltage of the power supply (V1), and the input voltage applied to the flapping-wing model (V2). Since the currents (I) applied to the resistor and flapping-wing model are the same [40], the current can be calculated as I=(V1-V2)/R1. Due to limitation in measuring the voltage applied to the motor (V3) inside the chamber, the resistance of the wire (R2) in the chamber was measured. Then, the voltage across the motor was calculated by V3=V2-IR2. Subsequently, the power consumption of the flapping-wing model can be calculated by multiplying the voltage V3 by the current I.
Comments 4: In lines 310-313, the authors mention that friction is primarily caused by higher flapping frequencies. However, it is important to note that additional friction may also result from the gravity component acting on the test setup. The authors should clarify whether gravity-induced friction was taken into account and if it might have impacted the results.
Response 4: As the flapping-wing system vibrates more, there are more chances to have contact between the rod and bearings, which may cause higher friction due to gravity. We have modified the explanation on friction as follows.
(See lines 321 to 324 on pages 10 to 11 in revised manuscript)
(Before)
The cycle-average lift coefficient remained consistent across all tilting angles and gravity conditions at approximately 1.08, which is an increase of approximately 6.2 % compared with 100 % air density. At a higher flapping frequency under a low air density, the increase was reasonable because of the higher friction caused by more vibrations. In the current case, the friction force supports the flapping-wing model such that it can hover at a lower flapping frequency, which results in a slightly higher cycle-average lift coefficient.
(After)
The cycle-average lift coefficient remained consistent across all tilting angles and gravity conditions at approximately 1.08, which is an increase of approximately 6.2 % compared with 100 % air density. At higher flapping frequencies under lower air densities (Tables 3 and 4), more vibration of the flapping-wing model may cause high friction between the rods and bearings, which results in an increase in the difference between measured and predicted hovering frequencies. In the current case, the friction force supports the flapping-wing model such that it can hover at a lower flapping frequency, which results in a slightly higher cycle-average lift coefficient.
Comments 5: In line 326, Table 1 appears again. Please correct the numbering of the tables to avoid confusion. Additionally, it is unclear whether the values in Table 1 represent averages across multiple trials or the results from a single experiment. Please clarify this point.
Response 5: In line 326, there is no Table 1. It is Table 5. Because there was no detailed explanation on the data in table 1, we explained how the data in table 1 were obtained reflecting your comment. The data in Table 1 were acquired by taking an average of three measurements. We included the explanation and standard deviations in the revised manuscript.
(See lines 232 to 235 on page 7 in the revised manuscript.)
In Table 1, “Tilting test (HSC)” means the flapping frequency measured by using a high-speed camera (FASTCAM Ultima APX, Photron, Japan), and “Load cell (Signal)” stands for the frequency processed from the force signal sensed by the load cell. The measurement data using load cell in Table 1 were acquired by taking an average of three measurements.
Lift (gf) |
Frequency (Hz) |
||
Prediction |
Measurement (Load cell) |
Tilting test |
Load cell |
11.70 () |
11.97 (+2.35 %) (SD=0.0449) |
14.21 |
14.16 (-0.35 %) (SD=0.0264) |
8.27 () |
8.49 (+2.69 %) (SD=0.0410) |
11.95 |
11.98 (+0.25 %) (SD=0.0273) |
5.85 () |
5.98 (+2.26 %) (SD=0.0595) |
10.07 |
10.08 (+0.10 %) (SD=0.0105) |
4.38 () |
4.49 (+2.61 %) (SD=0.0282) |
8.77 |
8.72 (+0.57 %) (SD=0.0182) |
Comments 6: All the supplementary videos currently provided are based on a purely lateral view, which makes it difficult to discern the differences in the experimental results. I recommend the following improvements:
- Provide additional views to capture different angles of the experiment.
- Use tracer particles to visualize airflow and flight patterns more effectively.
- Synchronize the presentation of experimental data for improved clarity and better demonstration of the effects observed in the trials.
Response 6: We appreciate your careful attention to the supplementary videos and your suggestions for improving the video quality.
- We included the lateral views only, because they could capture the hovering flight motion better than the others.
- The videos show only the moments of hovering due to the limitation of the memory size in the laptop. So, adding any tracer does not help to see the motion.
- Because we intend to show the hovering moment after takeoff and measure the flapping frequency at that moment, we do not have anything to synchronize.
Reviewer 2 Report
Comments and Suggestions for Authors
The manuscript investigates the prediction and measurement of hovering flapping frequency of flapping-wing aircraft under simulated low air density and gravity conditions. The study successfully demonstrates that the cycle-averaged lift coefficients remain relatively constant across different air density and gravity conditions, and it provides an analysis of energy consumption in thinner atmospheres. The job are significant for future flapping-wing flights on planets like Mars. The article is well-structured, with a clear experimental design and detailed data analysis. However, there is room for improvement, particularly in controlling experimental conditions and providing a deeper analysis of the results.
(1) The title reflects the content of the study, but could include keywords like "Mars" to highlight the practical application.
(2) The introduction discusses the advantages of flapping-wing aircraft but lacks a detailed analysis of their application prospects in space exploration. Additional literature or data support would be beneficial.
(3) The low-air density simulation method used in the experiment is well-established, but the analysis of potential sources of error is not sufficiently detailed. It would be beneficial to include a discussion on error control in the methods section.
(4) The article does not clearly point out the limitations of the study. It would be helpful to include an analysis of the limitations in the discussion section to help readers understand the scope of the study.
(5) The Reference No. 29 cited seem to be similar to the job in this paper. May be need a reasonable explanation.
(6) The conclusion summarizes the main findings but lacks specific suggestions for future research directions. It would be helpful to propose some concrete future research directions or improvements in the conclusion.
(7) Please carefully check and revise the references list according to the Instructions for Authors of this journal.
(8) The author should complete the language proofreading to eliminate any grammatical errors or unclear expressions.
Author Response
Reply to the reviewers' comments
Reviewer #2
We appreciate the Reviewer’s comments. All the comments were very useful to improve the quality of our paper. We have revised the manuscript reflecting most of the comments and hope that this revision fits the publication standards of Biomimetics.
Comments 1: The title reflects the content of the study, but could include keywords like "Mars" to highlight the practical application.
Response 1: Because we did not test for the Martian air density, we have not included the word, “Mars” in the title.
Comments 2: The introduction discusses the advantages of flapping-wing aircraft but lacks a detailed analysis of their application prospects in space exploration. Additional literature or data support would be beneficial.
Response 2: If there is air or gas on a planet, the advantage of flapping-wing aircraft on Earth does not change even though they are flying on the other planet. Furthermore, as mentioned in Introduction, there are only a few research reports on flapping flight under lower air density and gravity conditions can be found, which are all referred in this paper. Therefore, more literature or data are not available.
Comments 3: The low-air density simulation method used in the experiment is well-established, but the analysis of potential sources of error is not sufficiently detailed. It would be beneficial to include a discussion on error control in the methods section.
Response 3: The potential error may come from the pressure gage installed in the low-pressure chamber. We added the accuracy of the pressure gage in the revised manuscript.
(See lines 175 to 176 on page 5 in the revised manuscript.)
2.2.2. Low-air Density Condition
A chamber with adjustable pressure was used to simulate a low-density atmospheric environment. The chamber (0.73 m3) was the same as that used in [29]. The chamber was made of 15 mm thick polycarbonate (PC), and aluminum profiles (40 mm2) were installed inside to protect the PC walls. In this study, a flapping-wing flight experiment was conducted under the conditions of 25 %, 50 %, and 100 % air density at sea level. The potential error of the accuracy of air density can be ±1.5 % because it is the accuracy of the pressure gage (WS-110-Ï•60, WooShin, Korea).
Comments 4: The article does not clearly point out the limitations of the study. It would be helpful to include an analysis of the limitations in the discussion section to help readers understand the scope of the study.
Response 4: As pointed out in Discussion, the stand used to simulate the lower gravity condition cannot completely remove the effect of friction. The power prediction can be more accurate if we can add the power consumptions for lower air densities of 5 % and 10 %. We included this in the revised manuscript.
(See lines 394 to 402 on pages 12 to 13in the revised manuscript.)
(Before)
Figure 11B presents measured power data under various gravity conditions, fitted with a logarithmic function using the least-squares method for each air density. The logarithmic function was selected because it appropriately models the theoretical behavior in which the power consumption becomes infinite as the air density approaches zero, reflecting an impossible flight without air. This fitting function enables the prediction of power consumption for arbitrary combinations of air density and gravity conditions. Figure 11B also indicates that if the flight characteristics of the flapping-wing model remain consistent in the Martian environment, the power required for its hovering flight is estimated to be approximately 5.14 W. The flapping frequency necessary is predicted as 68.7 Hz for flight on Mars, according to the scaling Eq. (3). Thus, approximately 66 % more power is required to create a flapping frequency that is approximately five times higher on Mars than on Earth.
(After)
Figure 11B presents measured power data under various gravity conditions, fitted with a logarithmic function using the least-squares method for each air density. The logarithmic function was selected because it appropriately models theoretical behavior in which the power consumption becomes infinite as the air density approaches zero, reflecting an impossible flight without air. These fitting functions enable the prediction of power consumption for arbitrary combinations of air density and gravity conditions. Figure 11B also indicates that if the flight characteristics of the flapping-wing model remain consistent in the Martian environment, the power required for its hovering flight can be estimated by the fitting function (power = -1.059 ln (air density ratio) + 0.83). When the Martian air density ratio of 0.017 is substituted into the fitting function, the predicted power is approximately 5.14 W. The flapping frequency necessary is predicted as 68.7 Hz for flight on Mars, according to the scaling Eq. (3). Thus, approximately 66 % more power is required to create a flapping frequency that is approximately five times higher on Mars than on Earth. If we can add more measured power consumptions for lower air densities of 5 % and 10 % air density in the fitting curve, the power consumption for flapping flight on Mars can be more accurately predicted.
Comments 5: The Reference No. 29 cited seem to be similar to the job in this paper. May be need a reasonable explanation.
Response 5: The work in [29] is very different from the current research. As already mentioned in Introduction, in [29], the same voltages were applied to the flapping-wing model for all air density conditions to see how the flapping frequency and motion change in the low air density conditions for the same voltage input. On the other hand, in this study, we applied various voltages to make the flapping-wing model hover under low air densities and gravity conditions. Therefore, while the study in [29] focused on the change in wing flapping characteristics at low air density when the same voltage was applied, in this study, we investigated the change in wing flapping frequency and cycle- average lift coefficient when the flapping-wing model hovers under low air density and gravity conditions.
Comments 6: The conclusion summarizes the main findings but lacks specific suggestions for future research directions. It would be helpful to propose some concrete future research directions or improvements in the conclusion.
Response 6: We have revised the manuscript to include clear directions for future research and improvements based on your suggestions.
(See lines 427 to 433 on page 14 in the revised manuscript.)
(Before)
Decreasing air density requires the flapping-wing model to flap faster to achieve flight, resulting in higher power consumption. However, the reduced drag under low-air density conditions allows faster wing flapping at the same power level. From the power consumption data collected under various air density and gravity conditions, it is estimated that the power consumption required for a hovering flight on Mars is approximately 5.14 W, which is 66 % higher than the required for a hovering flight on Earth if the flight characteristics of the current flapping-wing model remain consistent under Martian conditions. Further precise assessment of friction under a low air density is necessary for a more accurate calculation of the cycle-average lift coefficient. This work has been extended to conduct experiments on hovering in a Martian environment.
(After)
Decreasing air density requires the flapping-wing model to flap faster to achieve flight, resulting in higher power consumption. However, the reduced drag under low-air density conditions allows faster wing flapping at the same power level. From the power consumption data collected under various air density and gravity conditions, it is estimated that the power consumption required for a hovering flight on Mars is approximately 5.14 W, which is 66 % higher than the required for a hovering flight on Earth if the flight characteristics of the current flapping-wing model remain consistent under Martian conditions.
To more accurately measure the hovering flapping frequency and calculate the cycle-average lift coefficient, we will suggest a way to reduce the effect of friction at higher flapping frequencies under low air density and gravity conditions in the future. We also plan to demonstrate hovering flapping flight of a flapping-wing model at lower air density conditions, which is closer to the Martian air density, and measure the corresponding power consumption for more accurate prediction of the power requirement for flapping flight on Mars.
Comments 7: Please carefully check and revise the references list according to the Instructions for Authors of this journal.
Response 7: We appreciate your careful attention. We have checked and revised the references list according to the Instructions.
Comments 8: The author should complete the language proofreading to eliminate any grammatical errors or unclear expressions.
Response 8: We have already had English correction service through a professional English polishing company before the initial submission. Despite that, we found some typos and checked English one more time.
Reviewer 3 Report
Comments and Suggestions for Authors
The review comments are detailed in the attached document.

Author Response
Reply to the reviewers' comments
Reviewer #3
We appreciate the Reviewer’s comments. All the comments were very useful to improve the quality of our paper. We have revised the manuscript reflecting most of the comments and hope that this revision fits the publication standards of Biomimetics.
Comments 1: Derivation Details of the Scaling Equation: While the current explanation of the scaling equation is informative, using the Buckingham Pi Theorem for further simplification could improve clarity and help readers better understand the physical basis of the equation. It is recommended that the authors provide a more detailed step-by-step derivation in the manuscript to strengthen the theoretical foundation.
Response 1: More details on the derivation are included in the revised manuscript.
(See lines 139 to 145 on page 4 in the revised manuscript)
(Before)
Here, the CL is the cycle-average lift coefficient. If the cycle-average lift coefficient remains constant across air density and gravitational acceleration, the flapping frequency required for hovering is inversely proportional to the square root of the air density and proportional to the square root of the gravitational acceleration [33]. Therefore, the scaling equation that predicts the flapping frequency required for hovering flight under specific air density and gravity conditions can be expressed by Eq. (3)
|
(3) |
|
|
|
|
where fE represents the flapping frequency required for flight on Earth, fA is the predicted flapping under different air density and gravity conditions, Rρ is the ratio of change in air density, and Rg is the ratio of change in gravity acceleration.
(After)
Here, the CL is the cycle-average lift coefficient. If the cycle-average lift coefficient remains constant across air density and gravitational acceleration, the flapping frequency required for hovering is inversely proportional to the square root of the air density and proportional to the square root of the gravitational acceleration. The ratio between required lift on Earth and another planet is expressed as follows:
|
(3) |
|
|
|
|
where the subscript E and A stand for Earth and another planet, respectively, and g means the gravity acceleration. Therefore, the scaling equation that predicts the flapping frequency required for hovering flight under specific air density and gravity conditions can be expressed by Eq. (4)
|
(4) |
|
|
|
|
From Eq. (4), it can be confirmed that a flapping-wing aircraft requires a faster flapping frequency as the air density decreases and can achieve flight with a slower flapping frequency as gravity decreases. After measuring the flapping frequency under a given set of air densities and gravity conditions, the measured flapping frequency was compared with the value predicted using Eq. (4). If the two frequencies are close, the cycle-average lift coefficient is considered constant.
Comments 2: Power Measurement Details: Although the power measurement methods have been described in other published works, it is commended that the authors include a more comprehensive description in this paper. This should cover the specifications of the measurement equipment, the voltage and current ranges, and the data acquisition process to ensure readers fully understand the power measurement procedures without needing to reference external sources.
Response 2: The power measurement is not very different from any power measurement of a device. Anyway, we included the details in the revised manuscript. The voltage range was from 3 V to 9 V and the current was dependent on voltage.
(See lines 257 to 264 on page 8 in the revised manuscript)
The power consumption of the flapping-wing model was measured at each hovering flapping frequency for a set of low-air-density and low-gravity conditions. The power measurement setup involved resistors, a power supply, and an oscilloscope (TDS 2024; Tektronix, USA), as shown in Figure 6. The oscilloscope measured the output voltage of the power supply (V1), and the input voltage applied to the flapping-wing system (V2). Since the currents (I) applied to the resistor and flapping-wing system are the same [40], the current can be calculated as I=(V1-V2)/R1. Due to limitation in measuring the voltage applied to the motor (V3) inside the chamber, the resistance of the wire (R2) in the chamber was measured. Then, the voltage across the motor was calculated by V3=V2-IR2. Subsequently, the power consumption of the flapping-wing system can be calculated by multiplying the voltage V3 by the current I.
Comments 3: Clarification of the 66% Power Increase Calculation: The conclusion states that the power required for hovering on Mars is 66% higher than on Earth. However, the calculation process leading to this figure is not clearly explained. It is recommended that the authors provide a more detailed breakdown of how this percentage was derived to ensure clarity, transparency, and reproducibility.
Response 3: The prediction was done by the curve fitting. The details are included in the revised manuscript.
(See lines 394 to 402 on pages 12 to 13 in the revised manuscript)
(Before)
Figure 11B presents measured power data under various gravity conditions, fitted with a logarithmic function using the least-squares method for each air density. The logarithmic function was selected because it appropriately models the theoretical behavior in which the power consumption becomes infinite as the air density approaches zero, reflecting an impossible flight without air. This fitting function enables the prediction of power consumption for arbitrary combinations of air density and gravity conditions. Figure 11B also indicates that if the flight characteristics of the flapping-wing model remain consistent in the Martian environment, the power required for its hovering flight is estimated to be approximately 5.14 W. The flapping frequency necessary is predicted as 68.7 Hz for flight on Mars, according to the scaling Eq. (3). Thus, approximately 66 % more power is required to create a flapping frequency that is approximately five times higher on Mars than on Earth.
(After)
Figure 11B presents measured power data under various gravity conditions, fitted with a logarithmic function using the least-squares method for each air density. The logarithmic function was selected because it appropriately models theoretical behavior in which the power consumption becomes infinite as the air density approaches zero, reflecting an impossible flight without air. These fitting functions enable the prediction of power consumption for arbitrary combinations of air density and gravity conditions. Figure 11B also indicates that if the flight characteristics of the flapping-wing model remain consistent in the Martian environment, the power required for its hovering flight can be estimated by the fitting function (power = -1.059 ln (air density ratio) + 0.83). When the Martian air density ratio of 0.017 is substituted into the fitting function, the predicted power is approximately 5.14 W. The flapping frequency necessary is predicted as 68.7 Hz for flight on Mars, according to the scaling Eq. (3). Thus, approximately 66 % more power is required to create a flapping frequency that is approximately five times higher on Mars than on Earth. If we can add more measured power consumptions for lower air densities of 5 % and 10 % air density in the fitting curve, the power consumption for flapping flight on Mars can be more accurately predicted.
Comments 4: Details on the Failure to Hover at 25% Air Density: Under 25% air density and a vertical orientation (1g gravity), the flapping-wing model was unable to take off. The authors should clarify whether this failure to take off persisted even with increased power input. Additionally, specifying the maximum power applied and the corresponding flapping frequency would help clarify the operational limitations and design challenges.
Response 4: Because the gear ratio is preset to 30.5:1, the maximum flapping frequency of the flapping-wing system is predetermined at about 23 Hz. Even though more power is applied to the motor, the maximum frequency is not increased higher than 23 Hz. The failure means that the flapping-wing system could not take off even though we applied voltages higher than the nominal voltage. We include the explanation in the revised manuscript.
(See lines 316 to 317 on page 10 in the revised manuscript)
(Before)
However, due to the significantly reduced air density, the generated lift was insufficient for takeoff at a tilting angle of 90° (1 g), as observed in Figure 9A.
(After)
However, due to the significantly reduced air density, the generated lift was insufficient for takeoff at a tilting angle of 90° (1 g), even when we applied voltages higher than the nominal voltage to the motor, as observed in Figure 9A.
Comments 5: Calculation of the Cycle-Average Lift Coefficient: Further clarification is needed on whether the cycle-average lift coefficient was calculated using the same gravity conditions and based on the experimentally measured flapping frequency or if it was derived purely from theoretical predictions using the scaling equation. Providing a clearer description of this calculation would ensure consistency and accuracy in data interpretation.
Response 5: The cycle-average lift coefficient was calculated based on the measured flapping frequency. We included the explanation in the revised manuscript
(See lines 287 to 288 on page 9 in the revised manuscript)
(Before)
3.2. Measured Flapping Frequency
The measured hovering flapping frequency, predicted frequency, and calculated cycle-average lift coefficient under different air densities and gravity conditions are summarized in Tables 2 through Table 4. In the first column of each table, the tilt angle simulates varying gravitational accelerations, as previously described. The lift in the second column lists the required lift for each gravity condition, which is computed by mg sin θ.
(After)
3.2. Measured Flapping Frequency
The measured hovering flapping frequency, predicted frequency, and cycle-average lift coefficient calculated based on the measured flapping frequency under different air densities and gravity conditions are summarized in Tables 2 through Table 4. In the first column of each table, the tilt angle simulates varying gravitational accelerations, as previously described. The lift in the second column lists the required lift for each gravity condition, which is computed by mg sin θ.
Round 2
Reviewer 1 Report
Comments and Suggestions for Authors
The author has basically answered all my questions, and I have no further inquiries.
Reviewer 2 Report
Comments and Suggestions for Authors
I think this manuscript has fit the publication standards of Biomimetics.